# Niche Differentiation between Two Sympatric *Cubitermes* Species (Isoptera, Termitidae, Cubitermitinae) Revealed by Stable C and N Isotopes

**DOI:** 10.3390/insects10020038

**Published:** 2019-02-01

**Authors:** Guy Josens, Solange Patricia Makatia Wango

**Affiliations:** 1Département de Biologie des Organismes, Université Libre de Bruxelles, Lab. Écologie Végétale et Biogéochimie, 50 av. Roosevelt, cp 244, B-1050 Brussels, Belgium; 2Département des Sciences de la vie, Faculté des Sciences, Université de Bangui, Bangui BP 908, Central African Republic; wasopa@yahoo.fr

**Keywords:** termite, humification, soil-feeding, trophic niche

## Abstract

Many African termite species are true soil-feeders: how can they coexist, sometimes with high densities? How do they separate their trophic niches? Preliminary results suggest that two coexisting *Cubitermes* species forage in different soil layers, and stable C and N isotopes show that they feed on different organic material. *Cubitermes aff. ugandensis* forages near the soil surface whereas *C. aff. sankurensis* forages in deeper layers; however, unexpectedly, the former shows a higher δ^15^N than the latter, highlighting, for the first time, a trophic niche differentiation between two sympatric true soil feeders bearing different enteric valve patterns.

## 1. Introduction

More than 60% of the termite species of the family Termitidae rely on soil-feeding [1]; among them, some are considered as wood–soil interface feeders and the others as true soil-feeders (“group III” and “group IV”, respectively, in [2]). The trophic niches of those species, besides the fact that they feed on organic material near the end of the humification gradient, are not well known.

The species of the mound-building genera *Thoracotermes* (four species) and *Cubitermes* (67 valid taxa listed in [3]) are known as true soil-feeders; these genera and some others of the same subfamily Cubitermitinae are good candidates for tackling the problem of trophic niche differentiation.

It has already been mentioned that two species of *Cubitermes* can frequently coexist in the same biotope: Williams [4] observed that *C. ugandensis* Fuller frequently shares the same biotope near Kampala, Uganda, with either *C. orthognathus* (Emerson) or *C. testaceus* Williams, suggesting that they might occupy somewhat different ecological niches. Bouillon & Vincke [5] suggested that the *Cubitermes’* diets might be linked with the morphologies of their enteric valves and, indeed, when looking at Williams’ data, *C. ugandensis* on the one hand has a valve of the *fungifaber* pattern whereas either *C. orthognathus* or *C. testaceus*, on the other hand, have valves of the *sankurensis* pattern [6]. The same kind of association was found at Bondoé, Republic of Central Africa: in some grassy savannas, two species previously identified as *C. ugandensis* and *C. sankurensis* Wasmann, coexist with high densities (more than 1200 mounds per hectare) [7].

Ji et al. [8] showed that *Cubitermes orthognathus* can partly assimilate radiolabeled synthetic humic acids but they did not mention any difference with *C. umbratus* Williams or *Thoracotermes macrothorax* Sjöstedt. Similarly, no difference was observed between *C. fungifaber* (Sjöstedt), *C. heghi* Sjöstedt, and *Th. macrothorax* in their stable isotope signatures [9].

This paper relates some preliminary results that were obtained at Bondoé, in November 2010; unfortunately, the civil war in the Republic of Central Africa prevented us, until now, to extend this research.

## 2. Material and Methods

The observations and sampling occurred in a grassy savannah at Bondoé, Republic of Central Africa (5°10′ N, 17°44′ E); the soil is shallow on a lateritic crust and can be occasionally flooded during the rainy season, but local small beaches offer deeper soil and many *Cubitermes* nests are located on those patches (more details in [7]).

Two nests, located on patches of relatively deep soil (about 30 cm), were chosen, the one built by *C. aff. ugandensis*, the other by *C. aff. sankurensis*. These species were identified as *C. ugandensis* and *C. sankurensis* in [7], but an ongoing revision and an ongoing phylogeny study of *Cubitermes* suggest that they belong to nearby, somewhat different species. The nest made by *C. aff. sankurensis* was 50 cm high, 25 cm in diameter, and had a single cap 45 cm in diameter; it is therefore evaluated to house about 35,000 individuals [10]. An annular trench, 25 cm deep, was dug around its base; the nest was examined 24 and 48 h later. The nest of *C. aff. ugandensis* was 30 cm high, 30 cm in diameter, and had two caps of about 40 cm in diameter; it is thus evaluated to contain about 30,000 individuals. This nest was also surrounded by an annular trench, 25 cm deep, and examined 24 and 48 h later.

Four other mounds, with a preference for recent nests, were identified to contain either *C. aff. sankurensis* (two nests) or *C. aff. ugandensis* (two nests). Recent mounds were chosen because they were expected to contain, each, only one species. Nevertheless, one of the *C. aff. sankurensis* mounds was somewhat older and appeared to house also a society of *Adaiphrotermes* sp.

A part of each of the nests was then sorted and samples of 20 to 30 workers, about 10 soldiers and 10 to 15 large larvae (and/or nymphs of alates) were collected and preserved in vials containing dry silica gel. Some nest fragments (about 10 g dry mass) were sampled and preserved the same way. The *Adaiphrotermes* sp. and the walls surrounding their galleries were also sampled. In the laboratory, the heads of the workers and the soldiers were cut off their bodies and preserved separately with dry silica gel. 

The samples were sent to the Kompetenzzentrum Stabile Isotope, Büsgen-Institut, Georg-August-Universität Göttingen for analysis of the C and N stable isotopes. The international standards for nitrogen and carbon are atmospheric nitrogen and Vienna Pee Dee Belemnite, respectively.

## 3. Results and Discussion

### 3.1. Depth of Foraging

The *C. aff. ugandensis* nest, around which a trench had been dug, showed, 24 and 48 h later, two covered galleries that were coming out of the nest at depths of 2 and 4 cm (therefore, near the soil surface) whereas the *C. aff. sankurensis* nest showed three covered galleries that were coming out of the nest at depths of 9, 20, and 21 cm, respectively. This preliminary result suggests that *C. aff. ugandensis* would forage near the surface of the soil whereas *C. aff. sankurensis* would forage in deeper layers. 

Even if it is based on two single observations, this is consistent with Williams’ observations near Kampala, Uganda: in that case, *C. ugandensis* coexists with *C. testaceus* (that has the same kind of enteric valve as *C. sankurensis*) and, there also, *C. ugandensis* tends to forage near the soil surface whereas *C. testaceus* does it “more randomly from the top to the lower parts of the topsoil” [4], p. 117.

It sounds logical that *C. aff. ugandensis*, whose workers and soldiers are much larger and possibly also much stronger than either *C. sankurensis* or *C. testaceus*, outcompetes the smaller species from the upper layers of soil which are richer in organic matter.

### 3.2. Stable Isotopes

#### 3.2.1. δ^13^C

In all the samples, the high δ^13^C values (between −13 and −10 in termites; between −14 to −12 in nest material) characterize soil-feeding and a grass food source [11,12], in this case, rather organic matter derived from grass (Figure 1).

In the worker heads, soldier heads, larvae, and alate nymphs, a tendency emerges: *C. aff. ugandensis* > *C. aff. sankurensis* > *Adaiphrotermes* sp., significantly different between *aff.. ugandensis* and *aff.. sankurensis* (*p* < 0.05, *t*-test). The same tendency is observed in the nest material, at least between *C. aff. ugandensis* and *C. aff. sankurensis*, but *Adaiphrotermes* established in a mound of *C. aff. sankurensis* did not change the δ^13^C value (Figure 1). Within each *Cubitermes* species, there is no difference between workers and soldiers (which are fed the same soil as workers) and their values have therefore been pooled in Figure 1; the larvae and alate nymphs (which are saliva-fed) display lower δ^13^C.

#### 3.2.2. δ^15^N

As expected for humivorous species, high δ^15^N values (between 7 and 12.5) were observed; actually between 9.5 and 12.5 in the true soil-feeders *Cubitermes* spp., and less high values (between 7 and 8.5) in the wood-soil interface feeder *Adaiphrotermes* sp.

In the worker and soldier heads, as well as larvae and alate nymphs, clear and consistent differences emerge: *C. aff. ugandensis* > *C. aff. sankurensis* >> *Adaiphrotermes* sp., significant between *aff.. ugandensis* and *aff.. sankurensis* (*p* < 0.01, *t*-test). The same tendency is observed in nest material, at least between *C. aff. ugandensis* and *C. aff. sankurensis*, but *Adaiphrotermes*, established in a mound of *C. aff. sankurensis*, shows the same δ^15^N value (Figure 1). The same tendency was observed in the abdomens whose δ^13^C and δ^15^N are lower than in the heads; however, these results are not shown in Figure 1 and difficult to interpret because we do not know which proportions of C or N correspond to termite tissues and to gut content.

In both *Cubitermes* species, there is no difference between workers and soldiers and their values have therefore been pooled in Figure 1; the larvae and alate nymphs display some lower δ^15^N. 

The difference between *Adaiphrotermes* sp. (feeding group III), on the one hand, and both *Cubitermes* species (feeding group IV), on the other hand, was expected, as it was already observed in Thailand and Cameroon for feeding groups III and IV [11,13]. However, since δ^15^N and δ^13^C are higher in deeper layers than in leaf litter and fermentation layers (more mineralized organic matter) [12,13,14,15], it sounds contradictory that *C. aff. ugandensis* (which forages in upper layers) shows higher δ^15^N (and probably also δ^13^C) in workers and soldiers, when compared with *C. aff. sankurensis* (which forages in deeper layers). 

## 4. Conclusions

The best possible explanation is that these species feed on different organic materials highlighting, for the first time, a trophic niche differentiation between two sympatric true soil feeders bearing two different enteric valve patterns. However, ^15^N and ^13^C signatures rely on several factors: exact nature of crude food, termite gut microbiome functions, ^15^N and ^13^C signatures of feeding saliva, etc. The “Bondoé bispecific model” provides in this scope promising future research perspectives.

## Figures and Tables

**Figure 1 insects-10-00038-f001:**
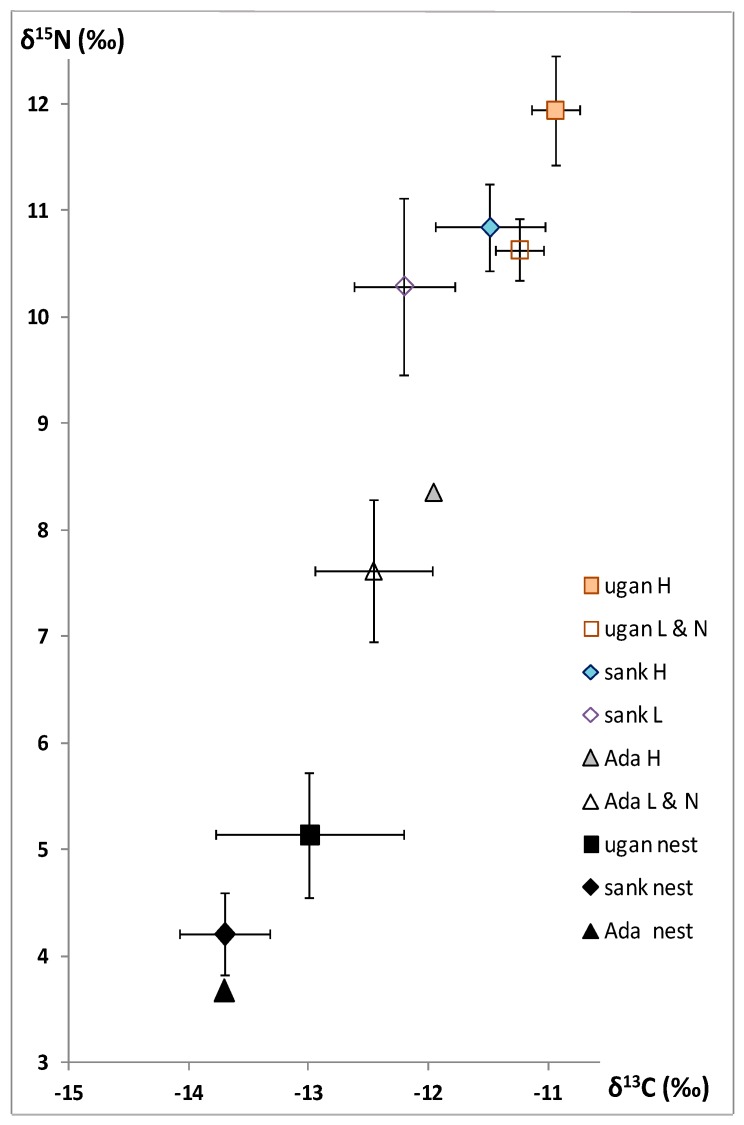
Plot of δ^15^N (‰) as function of δ^13^C (‰): ugan H = soldiers and workers heads of *C. aff. ugandensis*; ugan L & N = larvae and alate nymphs of *C. aff. ugandensis*; sank H = soldiers and workers heads of *C. aff. sankurensis*; sank L = larvae of *C. aff. sankurensis*; Ada H = workers heads of *Adaiphrotermes sp*; Ada L &N = larvae and alate nymphs of *C. Adaiphrotermes sp*; ugan nest = nest material from a mound of *C. aff. ugandensis*; sank nest = nest material from a mound of *C. aff. sankurensis*; Ada nest = nest material surrounding the galleries of *Adaiphrotermes* sp. (settled in a mound of *C. aff. sankurensis*). Data are expressed in mean ± SD, except for *Adaiphrotermes* heads and nest.

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
