# Peer review of "Niche Differentiation between Two Sympatric Cubitermes Species (Isoptera, Termitidae, Cubitermitinae) Revealed by Stable C and N Isotopes"

_insects, 2019, doi:10.3390/insects10020038_

Round 1

Reviewer 1 Report

The authors sought to use new technological approaches to establish niche preferences among two co-occuring termite species. Despite the limitations of unrest and war, the authors present results that support their hypothesis. However, i have some comments; some can be addressed and other might not be, but should be mentioned and explained.

1. The authors quantified carbon and nitrogen isotope signatures from nest, workers, soldiers and alates or larvae. Why werte dietary material in these mounds not collected? These are soil feeding termites so perhaps thats what the nest material refers to? If so then, but there is a variety of materials (plant and animal) potentially in the soil? The authors addressed this briefly in the results mentioning how the predominant material in the soil was grass but did not explain why the signature was reflective of soil feedin/grass feeding. They could explain more based on the references.

2. The authors should perhaps include a figure highlighting the difference in enrichment between termites and nest materials for both the 13C and 15N signatures. The authors should also report statistical results (either parametric or non) given sample restrictions?

3. Authors indicate that, 15N and 13C increases with soil depth, yet their results suggest the opposite in the co-occuring termite species. I believe some expansion of this contradictory result is required. The reason for this might be relevant. It could be that the top soil feeding termites may be feeding on materials enriched in 15N and 13C, relative to the bottom soil feeding termites? It may also be possible this may be due to gut microbial contributions between the two species, making the deeper dwelling termites have lighter isotope signatures relative to the top soil dwelling termites? I am aware that some of these questions cant be answered, but i don't believe that should limit the discussion. In fact I believe given the limitation of the study, it is important to try and put the results herein within some context.

4. The same suggestion applies to the observed differences in signature between workers and soldiers and larvae and alates. Why were the signatures lower in larvae relative to workers and soldiers?

5. Overall, i think the study has value, but should be expanded slightly to provide a more broader appeal. There are several interesting questions that remain unanswered , but this study sheds light on this ecological example and should provide how this fits in with termite ecology, termite gut microbiome compositions and functions and how this impacts isotope signature and concept and role in trophic functions.

Author Response

1. Cubitermes are indeed “soil-feeders”, however, random soil samples cannot be considered as their food: we are almost convinced that every species of humivorous termites selects its food but for the time being we do not know what it can be. More field work is requested to find out the precise places they feed on. The nest material has already been considered a proxy (in other papers) for dietary material; however, the nest material is a mixture of soil and faeces with ongoing microbial processes. This cannot be considered as equivalent to food. The predominant material in the soil derives from grass because we are in a grassy savannah (line 47), dominated by Loudetia sp, without any trees. The predominant material in the soil derives probably mainly from grass roots since annual bushfire destroys a large part of the epigeic vegetation.

2. The difference in enrichment between termites and nest material for both the 13C and 15N signatures can be seen on fig.1. It is difficult to interpret because of the unknown real origin of food and of nest material (that could be different). We made t-tests; however, we could not show all of them because most of them were suffering from pseudoreplication. We have nevertheless added in the paper some statistical information.

3. Indeed many factors could contribute to our result. Among them we think that grass root is probably the main origin of organic material in the soil; when the roots die, recent organic material simultaneously becomes available at several depths in the soil allowing an access to little enriched 15N even in deep layers. Unfortunately this is only speculative, more field work is requested: we hope we can continue this work once the political situation in Central Africa improves.

4. Actually, we expected to observe higher signatures in larvae/alates (than in workers) since they are saliva-fed and are in some way “predators” or “parasites” of the workers. A possible explanation of the opposite result would be that their food (“saliva”) has lower 15N and 13C signatures but this, again, is speculative.

5. OK, we have added some perspectives at the end of the article.

Reviewer 2 Report

Very interesting and innovative study that I took a lot of pleasure to read.

Besides the work of Tayasu and co., very few studies have considered the utilization of the isotopic signatures of termites for understanding their trophic niches. The question is especially relevant with soil feeding termites and this study clearly highlights the complexity and perhaps complementarity of this functional group. Like earthworms, can we now consider that we have oligo-, meso- and poly-humic soil feeding termites????????  

I just regret the fact that there is no information on the isotopic signature of soils and therefore the origin of the soil organic matter consumed by soil feeding termites. How does it change with soil depth? It would be nice to add this information if it is available. It also lacks a sentence explaining why only termite heads were analyzed – My first reaction would have been to analyze the whole individuals or at least to look at their abdomen. Don’t you think that this part of the body is more interesting to describe the origin of the soil that has been consumed?

I have a problem with the sentence L52-54 – “2 different species are different”???????

Finally, a small conclusion or a possible perspective to this study are perhaps missing. An interesting question would be (for me…) to determine why nest are clumped while termite species have very different signatures. Do termites homogenize the quality of soil organic matter????

Author Response

As far as we know, soil-feeding termites have the reputation to feed on organic-rich fractions of soil, they should therefore be considered as poly-humic soil-feeders. An oligo-humic regime (which would imply to swallow large amounts of soil with a long digestion time) seems incompatible with the small sizes of the termites. A meso-humic regime might be possible.

Only termite heads were analysed, as in other papers on humivorous termites. We actually also analysed the abdomens: their δ13C and δ15N are lower than in the heads; however, we did not include these results within the paper because we do not know which proportions of C or N correspond to termite tissues and to gut content. Sorry, this is a preliminary study. We hope we can continue this work once the political situation in Central Africa improves.

We (G. Josens and J. Deligne) have started a revision of the genus Cubitermes; a first article has been accepted and will be published soon (probably in January 2019) in the European Journal of Taxonomy (“Species groups in the genus Cubitermes (Isoptera: Termitidae) defined on the basis of enteric valve morphology”). In parallel some species, when fresh material is available, were sequenced (by our colleague S. Hellemans). So we could find that Cubitermes ugandensis from Central Africa, although very similar (morphologically) to the type of this species, should be considered as another species (and the same for C. sankurensis). These new species will be described later; for the time being we use C. aff ugandensis and C. aff sankurensis.

Termite nests (independently of their species) are clumped probably because the soil is shallow and/or located in a zone that floods during the rainy season. It can be imagined that two or three nests were initially built by chance near to each other. When old and abandoned these nests are weathered and the eroded soil spreads around, making the soil somewhat deeper. This place (with old ruined nests) are attracting for new settlers, etc. and after some generations, that makes kind of flat hills of deeper soil (= more food) and a relative shelter during the rainy season (surrounding soil temporarily flooded). Now another question is the “association” of two species in the clumps: their different signatures mean that they feed on different organic materials and that they are not competitors for the same food. We can even imagine that one of the species (through their faeces?) benefits the other but this is speculative. “Do termites homogenize the quality of soil organic matter?” Not any idea.